# Human T-cell leukemia virus type 1 may invalidate T-SPOT.*TB* assay results in rheumatoid arthritis patients: A retrospective case-control observational study

**Kunihiko Umekita** [1]*, **Yayoi Hashiba**[2], **Kosho Iwao**[1], **Chihiro Iwao**[1], **Masatoshi Kimura**[1], **Yumi Kariya**[1], **Kazuyoshi Kubo**[2], **Shunichi Miyauchi**[1], **Risa Kudou**[1], **Yuki Rikitake**[1], **Katoko Takajo**[1], **Takeshi Kawaguchi**[1], **Motohiro Matsuda**[1], **Ichiro Takajo**[1], **Eisuke Inoue**[3], **Toshihiko Hidaka**[2], **Akihiko Okayama**[1]

**1** Department of Rheumatology, Infectious Diseases and Laboratory Medicine, University of Miyazaki, Miyazaki, Japan, **2** Institute of Rheumatology, Zenjinkai Shimin-no-Mori Hospital, Miyazaki, Japan, **3** Showa University Research Administration Center, Showa University, Tokyo, Japan

\* kunihiko_umekita@med.miyazaki-u.ac.jp

**Data Availability Statement:** All relevant data are within the paper and its Supporting Information files.

## Abstract

### Background

CD4-positive T cells are the main target of human T-cell leukemia virus type 1 (HTLV-1). Interferon-γ release assays rely on the fact that T-lymphocytes release this cytokine when exposed to tuberculosis-specific antigens and are useful in testing for latent tuberculosis infection before initiating biologic therapy, such as anti-tumor necrosis factor agents. However, the reliability of interferon-γ release assays in detecting tuberculosis infection among HTLV-1-positive patients with rheumatoid arthritis (RA) remains unclear. The present study aimed to evaluate the use of the T-SPOT.*TB* assay in HTLV-1-positive RA patients.

### Methods

Overall, 29 HTLV-1-positive RA patients and 87 age- and sex-matched HTLV-1-negative RA patients (controls) were included from the HTLV-1 RA Miyazaki Cohort Study. Results of the T-SPOT.*TB* assay for latent tuberculosis infection screening were collected from medical records of patients.

### Results

Approximately 55% of the HTLV-1-positive RA patients showed invalid T-SPOT.*TB* assay results (odds ratio: 108, 95% confidence interval: 13.1–890, p < 0.0001) owing to a spot count of >10 in the negative controls. HTLV-1 proviral load values were significantly higher in patients with invalid results compared with those without invalid results (p = 0.003).

**Funding:** A part this work was supported by a grant from the Practical Research Project for Rare/Intractable Diseases of the Japan Agency for Medical Research and Development (Grant No. JP19ek0109356) to AO, a Health and Labor Sciences Research Grant on Rare and Intractable Diseases from the Ministry of Health, Labor and Welfare of Japan (Grant No. 19FC1007) to AO, and a Grant-in-Aid for Clinical Research from Miyazaki University Hospital to KU. The funders had no role in study design, data collection and analysis, decision to publish, or preparation of the manuscript.

**Competing interests:** The authors have declared that no competing interests exist.

## Conclusion

HTLV-1 infection affects T-SPOT.*TB* assay results in RA patients. Assay results in HTLV-1 endemic regions should be interpreted with caution when screening for latent tuberculosis infection before initiation of biologic therapy.

## Introduction

Standardization of anti-rheumatic treatment improves prognosis in patients with rheumatoid arthritis (RA). Tumor necrosis factor (TNF) antagonists are highly effective but associated with increased risk of tuberculosis (TB), mostly due to reactivation of a latent infection [1, 2]. Therefore, patients must be screened for latent TB infection (LTBI) before initiating anti-TNF agents. National recommendations for LTBI screening based on patient medical history, clinical examination, tuberculin skin testing (TST), and chest radiographs have been effective in reducing TB incidence [3]. However, the incidence of TB remains higher in patients receiving anti-TNF therapy compared with the general population [4, 5]. Furthermore, TST has well-known limitations: poor specificity due to cross-reactivity with environmental mycobacteria or bacillus Calmette–Guérin (BCG) vaccination [6] and poor sensitivity in immunocompromised patients [7, 8].

Interferon (IFN)-γ release assays (IGRAs) have been established as a screening test for LTBI. IGRAs are *in vitro* tests that rely on the rapid production of IFN-γ by CD4-positive effector memory or central memory T cells after stimulation with TB-specific antigens. In the general population, IGRAs are more effective than TST for diagnosing active TB infection or LTBI [9]. In 2010, the Centers for Disease Control and Prevention updated the guidelines for using IGRAs to detect TB infection [10]; IGRAs are recommended, because prior BCG vaccination does not lead to false-positive results. In clinical rheumatology, IGRAs are useful for diagnosing LTBI before the initiation of biologic therapy, such as anti-TNF agents [11]. Two different IGRAs for diagnosing TB infection—QuantiFERON-*TB* (QFT) and T-SPOT.*TB*—are currently available. Because whole blood is used in the QFT assay, the results may be affected by immunosuppressive therapies for RA [12]. On the other hand, the effect of immunosuppressive therapies may be less in T-SPOT.TB assay than in QFT because it is performed using isolated peripheral blood mononuclear cells (PBMCs); any immunosuppressive agents present are washed out [12]. Moreover, the T-SPOT.*TB* protocol is considerably easier to perform than the QFT protocol. The incidence of invalid results for the T-SPOT.*TB* assay is reportedly as low as 0.6% [13]. Therefore, this assay may be a useful tool for diagnosing LTBI in RA patients receiving immunosuppressive therapy.

Human T-cell leukemia virus type 1 (HTLV-1) is the causative agent of adult T-cell leukemia/lymphoma (ATL) and HTLV-1-associated myelopathy/tropical spastic paraparesis (HAM/TSP). HTLV-1 is endemic in Japan, where there are approximately 1 million HTLV-1 carriers [14]. CD4-positive T cells are the main target of the HTLV-1 virus. Some reports have found that the TST reaction in HTLV-1-positive individuals is attentuated compared with that in HTLV-1-negative individuals [15, 16]. These reports also suggest that HTLV-1 affects the adaptive immune response via HTLV-1-infected CD4-positive T cells. In addition, other reports have demonstrated that PBMCs isolated from HTLV-1-infected individuals automonously produce IFN-γ in cell culture conditions [17, 18]. However, the effect of HTLV-1 infection on TB IGRA results in RA patients remains unclear. Therefore, the present study aimed to evaluate the use of the T-SPOT.*TB* assay in HTLV-1-positive RA patients. In addition, the

association between IFN-γ-producing T cells and HTLV-1 proviral loads in HTLV-1-positive RA patients was examined. The present study demonstrated that HTLV-1 infection may invalidate T-SPOT.*TB* assay results in RA patients. Furthermore, HTLV-1-positive RA patients who have the high HTLV-1 PVL values tended to be showing invalid results for T-PSOT.*TB* assay.

## Materials and methods

### Study design and participants

The HTLV-1 RA Miyazaki Cohort Study was conducted from August 2012 to July 2019 at the Miyazaki University Hospital and Zenjinkai Shimin-no-Mori Hospital in the Miyazaki Prefecture, Japan [19]. The aim of this cohort study was to clarify the impact of HTLV-1 infection on the clinical features of RA patients and to investigate whether immunosuppressive therapies alter the risk factors associated with the development of ATL in HTLV-1-positive RA patients. A total of 858 RA patients were enrolled in this cohort. All participants were diagnosed with RA on the basis of the 1987 diagnostic criteria of the American College of Rheumatology (ACR) and screened for HTLV-1 infection [20]. Accordingly, 54 HTLV-1-positive RA patients were enrolled in this cohort. All RA patients were treated with anti-rheumatic drugs, such as methotrexate (MTX) and biologic agents, in accordance with RA treatment guidelines [21]. Written informed consent was obtained from all participants. These patients were expected to periodically visit the Miyazaki University and Zenjinkai Shimin-no-Mori Hospitals for clinical assessment and sample collection [19].

The participants of the present study were selected from this cohort. The inclusion criteria of this study as follows: HTLV-1-positive RA patients who underwent T-SPOT.*TB* assay (Oxford Immunotec, Oxford, UK) in this cohort from April 2012 to July 2019. The reasons for performing T-SPOT.*TB* assay was to detect LTBI before the initiation of treatment with biologic agents. In addition, the assay was performed when chest radiographs during anti-rheumatic treatment revealed findings suspicious for LTBI, such as pleural wall thickening, bronchial ectasia, and pleural effusion. According to this inclusion criteria, 29 of 54 HTLV-1-positive RA patients were enrolled into this study (Fig 1). Furthermore, 341 HTLV-1-negative RA patients had undergone T-SPOT.*TB* assay in this cohort during same observation period. Accordingly, 3 age- and sex-matched HTLV-1-negative RA patients from 341 HTLV-1-negative RA patients were selected as controls for each HTLV-1-positive RA patient; age was matched to within 5 years (Fig 1). To avoid sampling biases, the selection of the participants as controls was randomly performed from these 341 HTLV-1-negative RA patients based on sex and age, except for any other clinical information. Overall, 87 HTLV-1-negative RA patients were enrolled as controls in the present study. All clinical information evaluated during LTBI screening such as a previous TB history, TST results, T-SPOT.*TB* assay results, RA disease activity, anti-rheumatic regimen, white blood cell (WBC) count, and lymphocyte count were collected from medical records of these participants. The study protocol was approved by the research ethics committees of the Miyazaki University Hospital (approval no. O-0236) and Zenjinkai Shimin-no-Mori Hospital and followed the Ethical Guidelines for Medical and Health Research Involving Human Subjects.

### IFN-γ release assay

T-SPOT.*TB* assays (Oxford Immunotec, Oxford, UK) were performed by the CRC clinical laboratory company (Fukuoka, Japan), according to manufacturer's instructions [22]. Briefly, the assays were considered invalid if the negative control spot count was >10 or if the positive control spot count was <20 (low positive control). For valid tests, the result was obtained

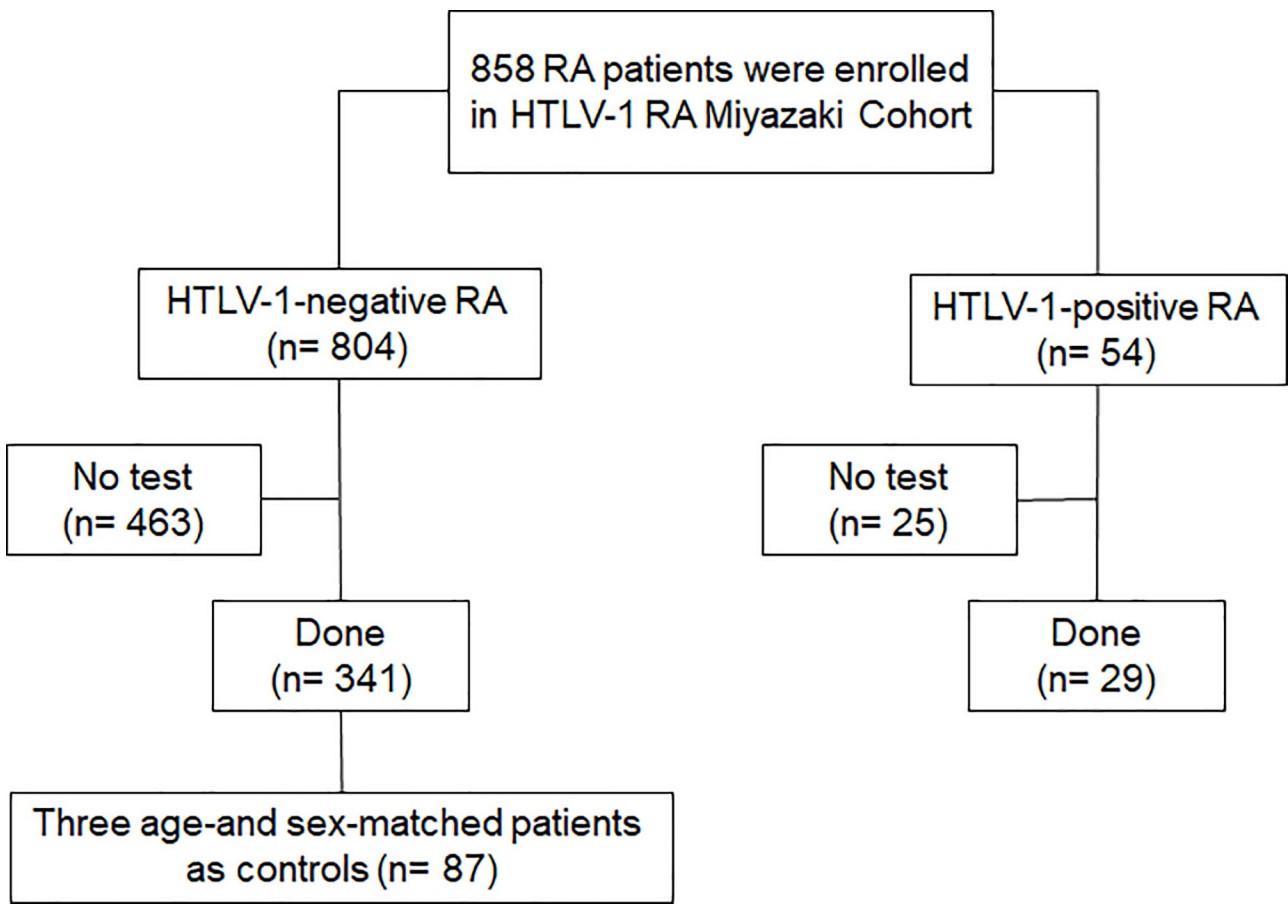

**Fig 1. The number of participants with RA who underwent T-SPOT.*TB* in HTLV-1 RA Miyazaki cohort.** A total of 29 HTLV-1-positive RA patients were evaluated for latent *Mycobacterium tuberculosis* infection using the T-SPOT.*TB* assay. Further, 341 HTLV-1-negative RA patients underwent T-SPOT.*TB* assay in this cohort. Accordingly, 3 age- and sex-matched HTLV-1-negative RA patients were selected as controls for each HTLV-1-positive RA patient. Finally, 29 HTLV-1-positive RA and 87 HTLV-1-negative RA patients were enrolled in this study.

(positive, negative, or borderline) by subtracting the spot count of the negative control from the highest spot count found between panels A (TB-specific antigen ESAT-6) and B (TB-specific antigen CFP-10). In this assay, when high background staining occurs hindering the discrimination of the spots from the background, the results were considered invalid. Therefore, if negative control spot count was >10 spots, the T-SPOT.*TB* assay result was considered invalid regardless of the spot count of both panel A and panel B.

## Clinical assessment of RA

Clinical information was collected either immediately before or after performing the T-SPOT. *TB* assay. RA disease activity was assessed using the 28-joint Disease Activity Score (DAS28), erythrocyte sedimentation rate (ESR), and the Clinical Disease Activity Index (CDAI) [23].

## HTLV-1 proviral load

HTLV-1 proviral load (PVL) was determined using peripheral blood samples obtained from HTLV-1-positive RA patients during follow-up clinical visits [19]. In HTLV-1-positive RA patients, DNA was purified from WBCs using the QIAamp Blood DNA Midi Kit (Qiagen, Hilden, Germany) and concentrated to 0.5 μg/μL via ethanol precipitation. Real-time PCR was

performed to measure the HTLV-1 *pX* region and the human albumin gene using the Light Cycler 2.0 thermal cycler (Roche Diagnostics, Mannheim, Germany) [24]. HTLV-1 PVL values in PBMCs were measured in duplicate, and the number of copies per 100 WBCs was calculated: a value of 4.0 copies per 100 PBMCs, which is a known risk factor for ATL development, corresponds to approximately 1.6 copies per 100 WBCs [25]. The HTLV-1 PVL values have been stored in the Miyazaki HTLV-1 RA Cohort database [19]. HTLV-1 PVL values that were measured immediately before or after performing the T-SPOT.*TB* assay were used in the present study, considering that anti-rheumatic treatments do not affect HTLV-1 PVL values in HTLV-1-positive RA patients [19].

## Statistical analysis

Statistical analyses were performed using EZR, R version 3.6.1. For describing patietns chracteristics, median with interquartile range (IQR) and count with proportion were used for continuous and categorical data, respectively. Group comparison between HTLV-1-negative and HTLV-1-positive RA patients was conducted using logistic regression model for age, the number of WBC and lymphocytes, inflammatory biomarker levels, DAS28, and CDAI. The sex ratio; rate of TST negativity; rate of rheumatoid factor (RF) and anti-citrullinated protein antibody (ACPA) positivity; and rate of corticosteroid, MTX, and biologic agent use between HTLV-1-negative and HTLV-1-positive RA patients were compared using logistic regression model. The HTLV-1 PVL values in HTLV-1-positive RA patients were compared between patients with negative and invalid results in the T-SPOT.*TB* assay using the Mann–Whitney test. Further, the invalid results of T-SPOT.*TB* assay were evaluated for HTLV-1-positive and negative group by odds ratio. $P < 0.05$ was considered significant.

## Results

### Characteristics of HTLV-1-negative and HTLV-1-positive RA patients

Table 1 shows the baseline clinical characteristics of the HTLV-1-negative and HTLV-1-positive RA patients enrolled in this study. There was no difference between the groups with respect to median age, sex ratio, seroprevalence of ACPA/RF, WBC and lymphocyte counts, CRP and ESR levels, and DAS28 and CDAI values. According to the European League Against Rheumatism (EULAR) improvement criteria, the median DAS28 for both groups indicated low RA activity. All patients were treated with disease-modifying anti-rheumatic drugs (DMARDs), including MTX and/or biologic agents, according to the ACR/EULAR guidelines [26]. The proportion of patients receiving corticosteroids and DMARDs excluding MTX did not differ between HTLV-1-negative and HTLV-1-positive RA patients. The rate of MTX use, median dosage of MTX, and proportion of patients using biologic agents did not differ between the groups. Both previous TB history and TST results were reviewed from medical records of the HTLV-1-positive and HTLV-1-negative RA patients. There wa no TB history in the HTLV-1-positive RA patients. In HTLV-1-negative RA patients, one of the 87 patients had a history of TB infection. Regarding TST results, 46 HTLV-1-negative and 13-HTLV-1-positive RA patients had undergone TST during the observation period. The rate of TST negativity tended to be high in HTLV-1-positive RA patients than in HTLV-1-negative RA patients (77% vs.67%, $P = 0.51$).

### T-SPOT.*TB* assay results in HTLV-1-negative and HTLV-1-positive RA patients

Table 2 shows the T-SPOT.*TB* assay results in the HTLV-1-negative and HTLV-1-positive RA patients. Overall, 55% of the HTLV-1-positive RA patients showed invalid results compared

**Table 1. Characteristics of HTLV-1-negative and positive patients with rheumatoid arthritis.**

| | HTLV-1-negative (n = 87) | HTLV-1-positive (n = 29) | P value |
|---|---|---|---|
| Age, years (IQR) | 70 (9.5) | 70 (9) | 0.94 |
| Female, no. (%) | 69 (79) | 23 (79) | - |
| Positive for RF, no. (%) [a] | 66/84 (78.5) | 20/28 (71.4) | 0.44 |
| Positive for ACPA, no. (%) [b] | 63/79 (79.7) | 14/20 (70) | 0.35 |
| CRP (mg/dL) (IQR) | 0.18 (1.4) | 0.19 (1.1) | 0.96 |
| ESR (mm/60 min) (IQR) | 28 (34) | 23 (33) | 0.17 |
| DAS28 (IQR) [c] | 3.11 (2.1) | 2.93 (1.7) | 0.61 |
| CDAI (IQR) [d] | 5.0 (11) | 6.6 (9.6) | 0.93 |
| White blood cell count (/μL) (IQR) | 5,654 (2,853) | 5,821 (3,024) | 0.74 |
| Lymphocyte count (/μL) (IQR) | 1,372 (598) | 1,648 (1,053) | 0.17 |
| Patients treated with corticosteroids, no. (%) | 28 (32.2) | 14 (48) | 0.12 |
| Corticosteroid dosage (mg/day) (IQR) [e] | 3.0 (3.0) | 2.75 (3.0) | 0.29 |
| Patients treated with conventional DMARDs (excluding MTX), no. (%) | 8 (9.2) | 5 (17.2) | 0.99 |
| Patients treated with MTX, no. (%) | 48 (55.1) | 12 (41.4) | 0.20 |
| MTX dosage (mg/week) (IQR) | 8.0 (4.0) | 8.0 (4.5) | 0.38 |
| Patients treated with biologic agents, no. (%) | 57 (65.5) | 15 (51.7) | 0.18 |
| Monotherapy (without MTX), no. (%) | 30 /57 (52.6) | 9 / 15 (60) | 0.61 |
| Combination with MTX, no (%) | 27/ 57 (47.4) | 6 / 15 (40) | 0.61 |

Values are expressed as medians with interquartile range (IQR). Percentages (%) are calculated based on total number of patients in each group unless indicated otherwise. CRP, C-reactive protein; ESR, erythrosedimentation rate; DAS28, 28-Joint Disease Activity Score; CDAI, Clinical Disease Activity Index; DMARDs, disease-modifying anti-rheumatic drugs; MTX, methotrexate.

[a] Data available in 84 and 28 patients of the HTLV-1-negative and HTLV-1-positive RA groups, respectively.

[b] Data available in 79 and 20 patients of the HTLV-1-negative and HTLV-1-positive RA groups, respectively.

[c] Data available in 87 and 26 patients of the HTLV-1-negative and HTLV-1-positive RA groups, respectively.

[d] Data available in 87 and 25 patients of the HTLV-1-negative and HTLV-1-positive RA groups, respectively.

[e] Prednisolone equivalent

with none of the HTLV-1-negative RA patients (odds ratio, 108; p < 0.0001). The cause of invalidity in the HTLV-1-positive RA patients was a spot count of >10 in the negative controls (Table 3). The median IFN-γ spot count in the negative control panels was 29.5 among the HTLV-1-positive RA patients with invalid results. The median IFN-γ spot count in the positive control panels of the HTLV-1-positive RA patients did not differ between those with invalid test results and those with negative test results (322 vs. 325, p = 0.93). There was no difference

**Table 2. T-SPOT.*TB* assay results in HTLV-1-negative and HTLV-1-positive patients with rheumatoid arthritis.**

| | HTLV-1-negative (n = 87) | HTLV-1-positive (n = 29) |
|---|---|---|
| Negative, n (%) | 81 (93.1) | 12 (41.4) |
| Positive, n (%) | 4 (4.7) | 0 |
| Borderline, n (%) | 1 (1.1) | 1 (3.5) |
| Invalid, n (%) | 1 (1.1) | 16 (55.1)* |

HTLV-1: human T-cell leukemia virus type 1.

*: P < 0.0001 Odds ratio[※]: 108 (95%CI: 13.1, 890).

※: Odds ratio of patients with HTLV-1 positive RA who have an invalid T-SPOT.*TB* test result to patients with negative RA.

**Table 3. IFN-γpositive spot counts in the T-SPOT.*TB* assay panels and HTLV-1 proviral load in HTLV-1-positive patients with rheumatoid arthritis.**

| | | PVL (copies/100 WBCs) | Spot count in T-SPOT.*TB* assay panels | | | | T-SPOT.*TB* assay result |
|---|---|---|---|---|---|---|---|
| | | | ESAT-6 | CFP10 | Negative control | Positive control | |
| Case | 1 | 15.12 | * | * | 18 | 240 | Invalid |
| Case | 2 | 4.43 | * | * | 16 | 346 | Invalid |
| Case | 3 | 3.89 | * | * | 65 | 489 | Invalid |
| Case | 4 | 3.84 | * | * | 18 | 467 | Invalid |
| Case | 5 | 2.95 | * | * | 13 | 307 | Invalid |
| Case | 6 | 2.88 | * | * | 86 | 338 | Invalid |
| Case | 7 | 2.83 | * | * | 66 | 219 | Invalid |
| Case | 8 | 2.6 | * | * | 102 | 452 | Invalid |
| Case | 9 | 2.45 | * | * | 28 | 254 | Invalid |
| Case | 10 | 2.03 | * | * | 17 | 180 | Invalid |
| Case | 11 | 2.03 | * | * | 11 | 258 | Invalid |
| Case | 12 | 1.54 | * | * | 185 | 615 | Invalid |
| Case | 13 | 1.5 | * | * | 14 | 299 | Invalid |
| Case | 14 | 0.89 | * | * | 93 | 365 | Invalid |
| Case | 15 | 0.46 | * | * | 38 | 303 | Invalid |
| Case | 16 | 0.19 | * | * | 31 | 431 | Invalid |
| Case | 17 | 8.48 | 0 | 0 | 0 | 445 | Negative |
| Case | 18 | 3.64 | 0 | 0 | 0 | 133 | Negative |
| Case | 19 | 1.42 | 1 | 0 | 1 | 396 | Negative |
| Case | 20 | 0.49 | 1 | 0 | 0 | 300 | Negative |
| Case | 21 | 0.37 | 0 | 0 | 1 | 208 | Negative |
| Case | 22 | 0.33 | 0 | 0 | 4 | 277 | Negative |
| Case | 23 | 0.2 | 0 | 0 | 3 | 203 | Negative |
| Case | 24 | 0.19 | 1 | 2 | 0 | 345 | Negative |
| Case | 25 | 0.1 | 0 | 0 | 0 | 403 | Negative |
| Case | 26 | 0.1 | 0 | 0 | 4 | 305 | Negative |
| Case | 27 | 0.04 | 0 | 0 | 1 | 670 | Negative |
| Case | 28 | 0.02 | 0 | 0 | 0 | 608 | Negative |
| Case | 29 | 2.61 | 7 | 0 | 2 | 128 | Borderline |

WBCs, white blood cells; PVL, proviral load; ESAT-6, early secretary antigen target 6; CFP10: culture filtrate protein 10

*, unevaluation.

in DMARD dosage or RA disease activity between the HTLV-1-positive RA patients with invalid results and those with negative results (S1 Table).

## HTLV-1 PVL in patients with invalid and valid T-SPOT.*TB* assay results

The median HTLV-1 PVL for the 29 HTLV-1-positive RA patients was 1.54 copies per 100 WBCs (IQR, 2.55). The median HTLV-1 PVL value was significantly higher in the 16 patients who showed invalid T-SPOT.*TB* assay results than in the 12 patients who showed conclusive negative results (2.52 vs. 0.33 copies/100 WBCs; p = 0.003) (Fig 2).

## Discussion

This is the first report demonstrating that HTLV-1 infection affects T-SPOT.*TB* assay results in RA patients, suggesting that the T-SPOT.*TB* assay may not be a reliable LTBI screening tool in approximately half of HTLV-1-positive RA patients.

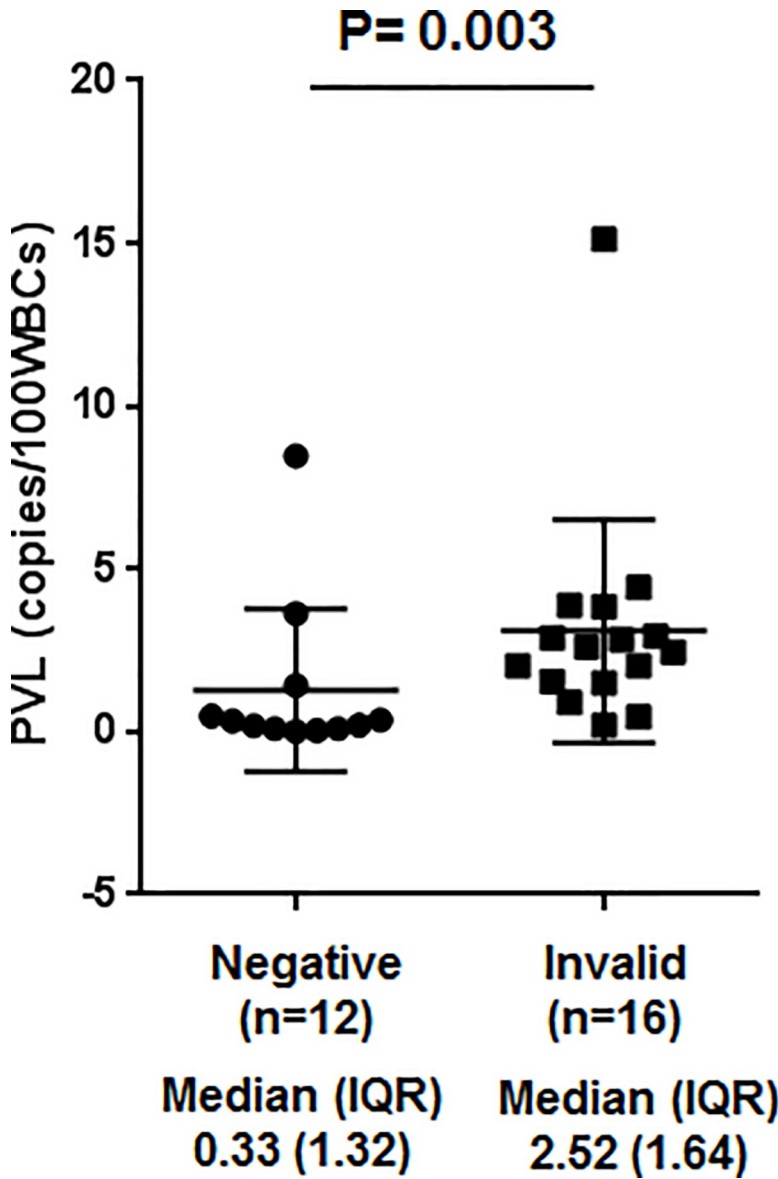

**Fig 2. Human T-cell leukemia virus type 1 (HTLV-1) proviral load (PVL) in HTLV-1-positive RA patients who showed negative (n = 12) or invalid (n = 16) T-SPOT.*TB* assay results.** The y-axis indicates HTLV-1 PVL in copies per 100 white blood cells (WBCs). Medians with interquartile range are shown.

In routine practice with regard to the management of rheumatic diseases, LTBI has been evaluated by IGRAs as well as radiologic examinations using chest high resolution computed tomography (HRCT). Additionally, it is important to determine history of previous TB infection before the administration of immunosuppressive therapy. In the present study, based on history of TB infection, T-SPOT.*TB* results, and HRCT findings, 19 HTLV-1-negative and 3 HTLV-1-positive RA patients were treated with isoniazid as a prophylactic regimen of TB reactivation before the initiation of antirheumatic biologic therapy. Although there was no history of previous TB infection in the HTLV-1-positive RA patients, 3 HTLV-1-positive RA patients had suspicious findings of LTBI in chest-HRCT and were administrated isoniazid. Among of them, 2 HTLV-1-positive RA patients showed an invalid T-SPOT.*TB* assay. In 19

HTLV-1-negative RA patients treated with isoniazid, 1 HTLV-1-negative RA patients had a history of TB infection and showed positive result for T-SPOT.*TB* assay. Further, 18 HTLV-1-negative RA patients had suspicious findings of LTBI in chest-HRCT and were administrated isoniazid. The invalid result of T-SPOT.*TB* assay is reportedly as low as 0.6% [13]. Therefore, this assay was recommended as a reliable tool to diagnose LTBI in RA patients receiving antirheumatic therapies including biologic agents. However, our result suggested that HTLV-1 infection may invalidate the T-SPOT.*TB* assay in HTLV-1-positive RA patients. In clinical practice, for the management of HTLV-1-positive RA patients, it is necessary to perform T-SPOT.*TB* assay as well as radiologic examinations for LTBI screening. In addition, if invalid result of T-SPOT.*TB* assay was observed in RA patients in HTLV-1 endemic areas, it would be preferable to consider the assessment of HTLV-1 infection.

Some previous reports have suggested that HTLV-1 infection may attenuate the response to purified protein derivative (PPD) of TB. The proportion of HTLV-1 carriers with low response to PPD of TB reportedly ranges from 65% to 70% [15, 16]. In the present study, the rate of TST negativity was 77% in HTLV-1-positive RA patients, which was high compared with those in previous reports [15, 16]. The reaction to PPD reportedly diminishes with advanced age. One of reasons for this tendency was considered that the median patient age in the present study was high compared with that in a previous report (70 vs. 63, respectively) [15, 16]. Moreover, the rate of TST negativity tended to be higher in HTLV-1-positive RA patients than that in HTLV-1-negative RA patients. Several studies have suggested that immunosuppressive treatments may affect the response to PPD. In patients with rheumatic diseases who have been treated with immunosuppressive agents, the efficacy of TST is influenced by low positive and negative predictive values [27, 28]. In the present study, the background of therapeutic regimens was similar between HTLV-1-negative and HTLV-1-positive RA patients. Therefore, it was considered that HTLV-1 infection may attenuate the TST reaction in HTLV-1-positive RA patients.

The cause of invalid results in the HTLV-1-positive patients was a high spot count in the negative controls (>10 spots) of the assay. A high spot count in the negative controls indicates the presence of autonomous IFN-γ producing cells without specific TB-antigen stimulation. HTLV-1-infected T cells often behave in a similar manner to T helper (Th) 1-like cells, which autonomously produce IFN-γ [17, 18]. HTLV-1-infected T cells reportedly show autonomous proliferation and produce inflammatory cytokines such as IL-6, TNF-alpha and IFN-γ [18]. HTLV-1 infection alters the original function of T cells via HTLV-1-associated proteins such as Tax and HBZ [18]. In HTLV-1-positive RA patients, these IFN-γ producing cells, which are suspected to be HTLV-1-infected T cells, may be responsible for the invalid T-SPOT.*TB* assay results.

In the present study, spot count in the negative control panels tended to be higher among HTLV-1-positive RA patients with high HTLV-1 PVL. Elevated PVL indicates an increased number of HTLV-1-infected T cells in carriers. In addition, high PVL is a risk factor for the development of ATL and HAM/TSP [29, 30]. Compared with healthy HTLV-1-negative patients or asymptomatic HTLV-1 carriers, HAM/TSP patients are characterized by elevated plasma levels of proinflammatory cytokines, such as IL-4, IL-6, IL-8, IFN-γ, and TNF-α [18]. IFN-γ-producing Th1-like HTLV-1-infected T cells play a crucial role in the pathogenesis of HAM/TSP. Our results suggest that the number of IFN-γ-producing HTLV-1-infected T cells is increased in HTLV-1-positive RA patients with high HTLV-1 PVL. However, the contribution of IFN-γ-producing, HTLV-1-infected T cells to RA pathogenesis remains unclear. Our previous studies indicated that HTLV-1-positive RA patients exhibit higher levels of inflammation and a poorer response to anti-TNF agents than HTLV-1-negative RA patients [31, 32].

In the future, an *ex vivo* study is necessary to investigate the pathologic role of IFN-γ-producing HTLV-1-infected T cells in HTLV-1-positive RA patients.

The present study has several limitations. First, the sample size was small. In addition, the clinical imformation was retrospectively collected from medical records; further, some items, such as autoantibodies associated with RA, disease activity scores, and TST results, were not completely assessed in this study. Because the inclusion criteria of this study was set as RA patients who underwent T-SPOT.*TB* assay for LTBI screening, the number of HTLV-1-positive RA patients who were enrolled can be small. Second, this study enrolled HTLV-1-positive RA patients but not asymptomatic HTLV-1 carriers. It remains unclear whether HTLV-1 infection invalidates T-SPOT.*TB* assay in asymptomatic HTLV-1 carriers. Therefore, the invalid results of the T-SPOT.*TB* assay due to HTLV-1 infection may not be generalizable for the population of asymptomatic HTLV-1 carriers. To clarify the impact of HTLV-1 infection in the T-SPOT.*TB* assay results, future studies including a large number of HTLV-1-positive RA patients as well as asymptomatic HTLV-1 carriers are required. Third, this study only evaluated the effect of HTLV-1 infection on T-SPOT.*TB* assay results. QFT results were not studied because only few HTLV-1 RA Miyazaki Cohort Study participants had undergone QFT assay. In the future, we plan to conduct a large-scale study involving both HTLV-1-positive RA patients and asymptomatic HTLV-1 carriers to further investigate the validity of both the T-SPOT.*TB* and QFT assays in these patients. Finally, the present study did not directly identify the specific cell type that was producing IFN-γ. Future studies should analyze IFN-γ production by HTLV-1-infected T cells using flow cytometry or *in vitro* cytokine arrays. In addition, the gene expression of HTLV-1-associated proteins, such as Tax and HBZ, should be investigated in IFN-γ-producing cells isolated from HTLV-1-positive RA patients.

In conclusion, invalid T-SPOT.*TB* assay results were observed in approximately half of HTLV-1-positive RA patients but not in HTLV-1-negative RA patients. Therefore, T-SPOT.*TB* assay results should be interpreted with caution when screening for LTBI in HTLV-1-positive RA patients. It remains unclear whether IFN-γ-producing cells in HTLV-1-positive RA patients have an effect on the inflammatory response in RA. Future investigation of the role played by these IFN-γ-producing cells may elucidate the pathogenesis of HTLV-1-associated diseases, such as HAM/TSP, as well as the interaction of inflammatory diseases, such as RA, with HTLV-1.

## Supporting information

**S1 Table. Characteristics of HTLV-1-positive rheumatoid arthritis patients with between negative and invalid results of T-SPOT.*TB* assay.**
(DOCX)

## Acknowledgments

We would like to thank Dr Yuki Hashikura and Ms Yuki Kaseda of the University of Miyazaki for their technical support in this work. We would also like to acknowledge Ms Yumiko Kai at the Institute of Rheumatology, Zenjinkai Shimin-no-Mori Hospital, for her help in data management. We would like to thank Enago (https://www.enago.jp/) for English language editing.

## Author Contributions

**Conceptualization:** Kunihiko Umekita.

**Data curation:** Yayoi Hashiba, Kosho Iwao, Chihiro Iwao, Masatoshi Kimura, Yumi Kariya, Kazuyoshi Kubo, Shunichi Miyauchi, Risa Kudou, Yuki Rikitake, Katoko Takajo, Takeshi Kawaguchi, Motohiro Matsuda, Ichiro Takajo, Toshihiko Hidaka.

**Formal analysis:** Kunihiko Umekita, Eisuke Inoue.

**Funding acquisition:** Kunihiko Umekita, Akihiko Okayama.

**Investigation:** Kunihiko Umekita, Yayoi Hashiba, Kosho Iwao.

**Methodology:** Kunihiko Umekita, Eisuke Inoue.

**Project administration:** Kunihiko Umekita.

**Supervision:** Toshihiko Hidaka, Akihiko Okayama.

**Validation:** Kunihiko Umekita, Chihiro Iwao, Yumi Kariya, Akihiko Okayama.

**Writing – original draft:** Kunihiko Umekita.

**Writing – review & editing:** Kunihiko Umekita, Akihiko Okayama.

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
