## [Decision Letter · Decision Letter 0]

20 Mar 2020

PONE-D-20-06768

Human T-cell leukemia virus type 1 may invalidate T-SPOT.TB assay results in rheumatoid arthritis patients: A retrospective observational study

PLOS ONE

Dear Dr Umekita,

Thank you for submitting your manuscript to PLOS ONE. After careful consideration, we feel that it has merit but does not fully meet PLOS ONE’s publication criteria as it currently stands. Therefore, we invite you to submit a revised version of the manuscript that addresses the points raised during the review process.

Specifically, both reviewers found some interests in this study, but pointed out a number of criticisms that require improvement or even amendment. I request the authors to respond fully to all comments made by reviewers. 

We would appreciate receiving your revised manuscript by May 04 2020 11:59PM. To enhance the reproducibility of your results, we recommend that if applicable you deposit your laboratory protocols in protocols.io, where a protocol can be assigned its own identifier (DOI) such that it can be cited independently in the future. For instructions see: http://journals.plos.org/plosone/s/submission-guidelines#loc-laboratory-protocols

We look forward to receiving your revised manuscript.

Kind regards,

Masataka Kuwana, MD, PhD

Academic Editor

PLOS ONE

Journal Requirements:

2. Please note that PLOS does not permit references to “data not shown.” Authors should provide the relevant data within the manuscript, the Supporting Information files, or in a public repository. If the data are not a core part of the research study being presented, we ask that authors remove any references to these data.

3. In your Methods section, please provide additional information about the participant recruitment method and the demographic details of your participants. Please ensure you have provided sufficient details to replicate the analyses such as: a) the recruitment date range (month and year), b) a description of any inclusion/exclusion criteria that were applied to participant recruitment, c) a description of how participants were recruited, and d) descriptions of where participants were recruited.

4. Please provide a sample size and power calculation in the Methods, or discuss the reasons for not performing one before study initiation.

Reviewers' comments:

Reviewer's Responses to Questions

**Comments to the Author**

1. Is the manuscript technically sound, and do the data support the conclusions?

Reviewer #1: Yes

Reviewer #2: Yes

2. Has the statistical analysis been performed appropriately and rigorously? 

Reviewer #1: Yes

Reviewer #2: Yes

3. Have the authors made all data underlying the findings in their manuscript fully available?

Reviewer #1: Yes

Reviewer #2: Yes

4. Is the manuscript presented in an intelligible fashion and written in standard English?

Reviewer #1: Yes

Reviewer #2: Yes

5. Review Comments to the Author

Reviewer #1: In this manuscript, the author presented HTLV-1 infection might affect T-SPOT.TB assay results in RA patients. They showed that T-SPOT. TB assay may not be a reliable LTBI screening tool in approximately half of the HTLV-1-positive patients. The author explained that the cause of invalid results in HTLV-1-positive patients was a high spot count in the negative controls of the assay. This phenomenon indicates the presence of autonomous IFN-γ producing cells without specific TB-antigen stimulation. The subject matter is impressive; however, much improvement can still be made to make it more unambiguous and more concise. The following are the points I raise after reading this manuscript.

Major points

1. In this study, 75 of 165 HTLV-1-negative and 29 of 55 HTLV-1-positive RA patients from 861 RA patients in the HTLV-1 RA cohort were enrolled. Please show the patient registration process flowchart for this study in the figure.

2. Paragraph on study design and participants, page 5, please specify the "initial study criteria," and in particular, the eligibility criteria and the sources and methods of selection of participants, according to the STROBE checklist.

3. T-Spot TB tests may also be positive in patients with a previous history of TB. What was the proportion of patients with a history of TB in the HTLV -1 positive and negative groups?

4. Even if the result of the T-spot TB test is invalid, it is necessary to evaluate some TB infection before the treatment of csDMARDs or biologic DMARDs. How did the author assess the 16 patients with an invalid T-spot TB test in the 29 HTLV -1 positive group for M. tuberculosis infection?

5. In the discussion section, please describe your efforts to address potential sources of bias, as per the STROBE checklist, addressing in particular confounding variables, lack of generalizability, selective reporting, post hoc analyses, and data dredging. In the paragraph on limitations (page 14-15)

Minor points

1. Anti-rheumatic treatment is misspelled on page 5 of line 16.

2. Please add a 95% confidence interval for the odds ratio of patients with HTLV-1 positive RA who have an invalid T-spot TB test result to patients with negative RA (Table 2).

Reviewer #2: Dear the author,

This article describes the tendency of invalid result in T-SPOT.TB assay among RA patients.

It seems important as well as interesting because using bDMARDs increases the possibility of symptomatic TB infection, and according to the manuscript physicians must pay attention how to interpret T-SPOT.TB assay from HTLV-1 positive RA patients.

However, there are a few minor issues that should be modified by the authors.

1. If the number of HTLV-1 positive patients to whom LTBI screening were carried out were twenty-nine, age- and sex- matching of HTLV-1 negative RA patients should be done with such 29 patients, not with 55 patients, even if the result as shown in Table 1 was suitable for continuing the study.

2. In page 5, line 16, “anti-fheumatic” should be replaced with “anti-rheumatic”.

3. To evaluate the appropriate comparison, WBC count and/or Lymphocyte count should be added in Table 1.

4. According to the result of T-SPOT.TB assay in HTLV-1 positive patients, spot count in the negative controls tends to >10 but that in EAST-6 and CFP10 are almost zero. Authors should explain the discrepancy. could the proteins eliminate the production of IFN-gamma from T cells in HTLV-1 positive patients?

5. How about a tuberculin reaction in HTLV-1 positive patients? Previous report should be cited in discussion page.

6. PLOS authors have the option to publish the peer review history of their article (what does this mean?). If published, this will include your full peer review and any attached files.

Reviewer #1: No

Reviewer #2: Yes: Kosaku Murakami

---

## [Author Response · Author response to Decision Letter 0]

27 Apr 2020

Respond to Reviewer’s comments.

Dear Editor and Reviewers,

Thank you for your careful review and the pertinent and insightful comments.

We have revised our manuscript following reviewer’s suggestions. We hope that our edits and the responses we provide below satisfactorily address all the issues and concerns you and the reviewers have noted.

Best regards,

Kunihiko Umekita

Reviewer #1: In this manuscript, the author presented HTLV-1 infection might affect T-SPOT.TB assay results in RA patients. They showed that T-SPOT. TB assay may not be a reliable LTBI screening tool in approximately half of the HTLV-1-positive patients. The author explained that the cause of invalid results in HTLV-1-positive patients was a high spot count in the negative controls of the assay. This phenomenon indicates the presence of autonomous IFN-γ producing cells without specific TB-antigen stimulation. The subject matter is impressive; however, much improvement can still be made to make it more unambiguous and more concise. The following are the points I raise after reading this manuscript.

Major points

1. In this study, 75 of 165 HTLV-1-negative and 29 of 55 HTLV-1-positive RA patients from 861 RA patients in the HTLV-1 RA cohort were enrolled. Please show the patient registration process flowchart for this study in the figure.

We thank you for your suggestion. In line with a comment from another reviewer, we changed the study design to a case control study. In addition, we invited a specialist for medical statistics in the present study. We revised the method of statistical analysis because of change the study design (Page 9, line 151 to line 159). We selected three age- and sex-matched patients with HTLV-1-negative RA for each patient with HTLV-1-positive RA as controls. Eighty-seven patients with HTLV-1-negative RA and 29 patients with HTLV-1-positive RA were enrolled in this study (Page 5, line 91 to Page 6, line 106). We presented the patient registration process flowchart in Figure 1.

2. Paragraph on study design and participants, page 5, please specify the "initial study criteria," and in particular, the eligibility criteria and the sources and methods of selection of participants, according to the STROBE checklist.

We described the initial study criteria, the eligibility criteria, and the sources and methods of patient selection according to the STROBE checklist (Page 5, line 91 to page 6, line 106).

3. T-Spot TB tests may also be positive in patients with a previous history of TB. What was the proportion of patients with a history of TB in the HTLV -1 positive and negative groups?

We appreciate your comment. We collected data on the history of TB infection from the patients’ medical records. No patients with HTLV-1-positive RA had a history of TB, compared with one patient with HTLV-1-negative RA. The positive result was obtained using the T-SPOT.TB assay. We have added this information in the Results (Page 10, Lines 171–174).

4. Even if the result of the T-spot TB test is invalid, it is necessary to evaluate some TB infection before the treatment of csDMARDs or biologic DMARDs. How did the author assess the 16 patients with an invalid T-spot TB test in the 29 HTLV -1 positive group for M. tuberculosis infection?

We thank you for your comment. In daily practice, we evaluate latent TB infection (LTBI) using IGRAs such as T-SPOT.TB and radiologic examinations such as high-resolution computed tomography (HRCT) of the chest. Additionally, it is important to assess the history of TB infection. Of the 16 patients with an invalid T-SPOT.TB assay result, two participants with suspicious findings of LTBI on chest HRCT were administrated prophylactic regimens targeting TB reactivation such as isoniazid before starting anti-rheumatic biologic therapy. In the present study, based on the history of TB infection, T-SPOT.TB result, and HRCT findings, 19 patients with HTLV-1-negative RA and 3 patients with HTLV-1-positive RA were treated with isoniazid as prophylaxis. We described this information in the Discussion (Page 16, Line 207 to Page 17, Line 225).

5. In the discussion section, please describe your efforts to address potential sources of bias, as per the STROBE checklist, addressing in particular confounding variables, lack of generalizability, selective reporting, post hoc analyses, and data dredging. In the paragraph on limitations (page 14-15)

We described the limitation according to the STROBE checklist (Page 19, Line 261 to Page 20, Line 271)

Minor points

1. Anti-rheumatic treatment is misspelled on page 5 of line 16.

We corrected the spelling of this word．

2. Please add a 95% confidence interval for the odds ratio of patients with HTLV-1 positive RA who have an invalid T-spot TB test result to patients with negative RA (Table 2).

We provided the 95% confidence interval in Table 2.

Reviewer #2: Dear the author,

This article describes the tendency of invalid result in T-SPOT.TB assay among RA patients.

It seems important as well as interesting because using bDMARDs increases the possibility of symptomatic TB infection, and according to the manuscript physicians must pay attention how to interpret T-SPOT.TB assay from HTLV-1 positive RA patients.

However, there are a few minor issues that should be modified by the authors.

1. If the number of HTLV-1 positive patients to whom LTBI screening were carried out were twenty-nine, age- and sex- matching of HTLV-1 negative RA patients should be done with such 29 patients, not with 55 patients, even if the result as shown in Table 1 was suitable for continuing the study.

We thank you for your suggestion. We changed the study design to a case control study according to your suggestion. Therefore, we invited a specialist for medical statistics because of change the study design. We described the protocol in the methods section (Page 5, Line 91 to Page 6, Line 106). In addition, we revised the method of statistical analysis (Page 9, line 151 to line 159). We presented the flowchart of patient selection as Figure 1 according to the suggestion of another reviewer.

2. In page 5, line 16, “anti-fheumatic” should be replaced with “anti-rheumatic”.

We corrected the spelling of this word.

3. To evaluate the appropriate comparison, WBC count and/or Lymphocyte count should be added in Table 1.

We added data for the WBC and lymphocyte counts in Table 1.

4. According to the result of T-SPOT.TB assay in HTLV-1 positive patients, spot count in the negative controls tends to >10 but that in EAST-6 and CFP10 are almost zero. Authors should explain the discrepancy. could the proteins eliminate the production of IFN-gamma from T cells in HTLV-1 positive patients?

We thank you for your comments. In this assay, when high background staining hinders the discrimination of the spots from the background, the results should be considered invalid. Therefore, if the negative control spot count exceeds 10, the T-SPOT.TB result should be considered invalid regardless of the spot counts in both panels A and B. Several clinical laboratory companies report spot counts as zero in both panels A and B in the T-SPOT-TB assay if they observe high negative control spot counts. In the present study, we initially used a count of zero to indicate unevaluable results in Table 3 in line with the reports of clinical laboratory companies. To avoid reader confusion, we would like to use asterisks (*) to indicate unevaluable results for both panels A and B in Table 3. We described the assessment of the T-SPOT.TB assay in detail in the methods section (Page 7, lines 122 - 127). In addition, we described the meaning of this symbol in Table 3. 

5. How about a tuberculin reaction in HTLV-1 positive patients? Previous report should be cited in discussion page.

In the present study, 46 patients with HTLV-1-negative RA and 13 patients with HTLV-1-positive RA underwent tuberculin skin tests (TSTs) during the observation period. The proportion of patients with negative TST results tended to be higher in the HTLV-1-positive RA group than in the HTLV-1-negative RA group (77% versus 67%, P = 0.73). We described this finding in the Results (Page 10, Line 174 to line 177).

Some previous reports suggested that HTLV-1 infection can attenuate the response to purified protein derivative (PPD) of Mycobacterium tuberculosis. The proportion of HTLV-1 carriers who had a low response to PPD was reported to range 65–70% (Ref. 15–16). 

The proportion of patients with negative TST results in the present study tended to be higher than those of previous studies. It is well known that the reaction to PPD diminishes with age. One reason for this tendency observed in the present study could be the higher median age of the participants compared with previous reports (70 years versus 63 years). 

Several studies suggested that immunosuppressive treatments can affect the response to PPD. In patients with rheumatic diseases who have been treated with immunosuppressive agents, TST is hampered by low positive and negative predictive values (Ref. 31–32). The background therapeutic regimen was nearly identical between the HTLV-1-negative and HTLV-1-positive RA groups in the present study. Therefore, it was considered that HTLV-1 infection can attenuate the TST result in patients with HTLV-1-positive RA. 

We describe these findings in the Discussion (Page 17, Line 226 to Page 18, Line 238).

---

## [Decision Letter · Decision Letter 1]

30 Apr 2020

Human T-cell leukemia virus type 1 may invalidate T-SPOT.TB assay results in rheumatoid arthritis patients: a retrospective case-control observational study

PONE-D-20-06768R1

Dear Dr. Umekita,

We are pleased to inform you that your manuscript has been judged scientifically suitable for publication and will be formally accepted for publication once it complies with all outstanding technical requirements.

With kind regards,

Masataka Kuwana, MD, PhD

Academic Editor

PLOS ONE

Additional Editor Comments (optional):

Reviewers' comments:

Reviewer's Responses to Questions

**Comments to the Author**

1. If the authors have adequately addressed your comments raised in a previous round of review and you feel that this manuscript is now acceptable for publication, you may indicate that here to bypass the “Comments to the Author” section, enter your conflict of interest statement in the “Confidential to Editor” section, and submit your "Accept" recommendation.

Reviewer #1: All comments have been addressed

Reviewer #2: All comments have been addressed

2. Is the manuscript technically sound, and do the data support the conclusions?

Reviewer #1: Yes

Reviewer #2: Yes

3. Has the statistical analysis been performed appropriately and rigorously? 

Reviewer #1: Yes

Reviewer #2: Yes

4. Have the authors made all data underlying the findings in their manuscript fully available?

Reviewer #1: Yes

Reviewer #2: Yes

5. Is the manuscript presented in an intelligible fashion and written in standard English?

Reviewer #1: Yes

Reviewer #2: Yes

6. Review Comments to the Author

Reviewer #1: The authors have satisfactorily responded to all my questions and made the necessary changes to the manuscript. So, I have no further comments in this manuscript.

Reviewer #2: Dear the author, The manuscript "Human T-cell leukemia virus type 1 may invalidate T-SPOT.TB assay results in rheumatoid arthritis patients: a retrospective case-control observational study" (PONE-D-20-06768R1) was properly edited according to the reviewers' comments.

7. PLOS authors have the option to publish the peer review history of their article (what does this mean?). If published, this will include your full peer review and any attached files.

Reviewer #1: No

Reviewer #2: Yes: Kosaku Murakami

---

## [Editor Report · Acceptance letter]

8 May 2020

PONE-D-20-06768R1 

Human T-cell leukemia virus type 1 may invalidate T-SPOT.*TB* assay results in rheumatoid arthritis patients: a retrospective case-control observational study 

Dear Dr. Umekita:

I am pleased to inform you that your manuscript has been deemed suitable for publication in PLOS ONE. Congratulations! Your manuscript is now with our production department. 

With kind regards,

on behalf of

Prof. Masataka Kuwana 

Academic Editor

PLOS ONE